# Haploinsufficiency of Adenomatous Polyposis Coli Coupled with Kirsten Rat Sarcoma Viral Oncogene Homologue Activation and P53 Loss Provokes High-Grade Glioblastoma Formation in Mice

**DOI:** 10.3390/cancers16051046

**Published:** 2024-03-04

**Authors:** Kuan-Te Fang, Chuan-Shiang Su, Jhoanna Jane Layos, Nga Yin Sadonna Lau, Kuang-Hung Cheng

**Affiliations:** 1Institute of Biomedical Sciences, National Sun Yat-Sen University, Kaohsiung 804, Taiwan; calvinfang@hotmail.com (K.-T.F.); m022050012@student.nsysu.edu.tw (C.-S.S.); d118020008@nsysu.edu.tw (J.J.L.); d118020009@student.nsysu.edu.tw (N.Y.S.L.); 2National Institute of Cancer Research, National Health Research Institutes, Tainan 704, Taiwan; 3Department of Medical Laboratory Science and Biotechnology, Kaohsiung Medical University, Kaohsiung 807, Taiwan

**Keywords:** animal models, glioblastoma multiforme (GBM), APC haploinsufficiency, giant cells

## Abstract

**Simple Summary:**

Glioblastoma multiforme (GBM) ranks as the most frequent form of primary malignant brain tumor. The prognosis for individuals diagnosed with this disease typically leads to a median survival of under two years. Here, we developed mouse models to better understand the genetic basis of GBM, a highly aggressive brain tumor. We found that certain genetic alterations, such as Kirsten rat sarcoma viral oncogene homologue (KRAS) activation and p53 deficiency, cooperate to initiate glioma tumorigenesis. Combining these alterations with adenomatous polyposis coli (APC) haploinsufficiency led to the rapid progression of GBM in the mice, resembling the human disease. These models are valuable for identifying early disease biomarkers and may offer insights for improving the diagnosis and treatment of this challenging brain tumor.

**Abstract:**

Glioblastoma multiforme (GBM) is the most common and deadly type of brain tumor originating from glial cells. Despite decades of clinical trials and research, there has been limited success in improving survival rates. However, molecular pathology studies have provided a detailed understanding of the genetic alterations associated with the formation and progression of glioblastoma—such as Kirsten rat sarcoma viral oncogene homolog (KRAS) signaling activation (5%), P53 mutations (25%), and adenomatous polyposis coli (APC) alterations (2%)—laying the groundwork for further investigation into the biological and biochemical basis of this malignancy. These analyses have been crucial in revealing the sequential appearance of specific genetic lesions at distinct histopathological stages during the development of GBM. To further explore the pathogenesis and progression of glioblastoma, here, we developed the glial-fibrillary-acidic-protein (GFAP)-Cre-driven mouse model and demonstrated that activated KRAS and p53 deficiencies play distinct and cooperative roles in initiating glioma tumorigenesis. Additionally, the combination of APC haploinsufficiency with mutant Kras activation and p53 deletion resulted in the rapid progression of GBM, characterized by perivascular inflammation, large necrotic areas, and multinucleated giant cells. Consequently, our GBM models have proven to be invaluable resources for identifying early disease biomarkers in glioblastoma, as they closely mimic the human disease. The insights gained from these models may pave the way for potential advancements in the diagnosis and treatment of this challenging brain tumor.

## 1. Introduction

Glioma, a primary tumor of glial cell origin, is the most common intracranial neoplasm, accounting for more than 80% of brain tumors. Glioblastoma multiforme (GBM) is the most malignant brain tumor and is linked to one of the worst 5-year survival rates among human cancers [1]. Despite intensive treatment, most patients diagnosed with GBM will die within the first two years. However, in recent years, the survival rate of patients with GBM has improved from an average of 10 to 14 months due to improvements in the standard of care [2,3]. Of note, men have been found to have a higher incidence of primary GBM than women, with a ratio of 1.58:1 [4]. Meanwhile, the World Health Organization (WHO) classifies glioma stages as grades II to IV based on tumor histological and immunohistochemical (IHC) features, ranging from low-grade to high-grade astrocytomas [2,5,6]. Primary GBM, which progresses rapidly and has an absence of precursor lesions, constitutes up to 95% of GBM cases and stands as the most prevalent malignant primary brain tumor, comprising 54% of all gliomas [7,8]. On the other hand, a secondary glioblastoma progresses from a low-grade diffuse astrocytoma or anaplastic astrocytoma and is often diagnosed in a younger population, with a median age of 45 years old, and it is more common in women [9].

Similar to other human malignancies, the molecular progression of gliomas involves an accumulation of genetic and epigenetic alterations that results in the loss of tumor suppressor genes or the activation of oncogenic pathways [10,11,12]. Recent genetic studies have shown that deletions or mutations in certain genes can lead to the formation of GBM. For instance, primary GBM is characterized by epidermal growth factor receptor (EGFR) gene amplification and mutations, the loss of heterozygosity (LOH) of chromosome 10q containing phosphatase and tensin homolog (PTEN), the overexpression of mouse double minute 2 (MDM2), and the deletion of the p16 tumor suppressor gene [13]. Notably, Kras activation in GBM presents a significant molecular event that plays a pivotal role in the development of this aggressive brain tumor [14,15,16]. The Kras gene encodes a protein called Kirsten rat sarcoma viral oncogene homolog (KRAS), which is a member of the Ras family of small GTPases. In GBM, Kras activation is frequently triggered by mutations in the KRAS gene, resulting in the continuous activation of the KRAS signaling cascades. This persistent Kras activation leads to the continuous activation of downstream signaling pathways, including the mitogen-activated protein kinase (MAPK) and phosphoinositide 3-kinase (PI3K) pathways, which regulate cell proliferation and survival [17]. The abnormal activation of Kras in GBM is believed to contribute to increased cell proliferation, resistance to cell death, and enhanced tumor growth [18,19]. On the other hand, P53 has been known to function as a tumor suppressor in regulating cell growth, DNA repair, and apoptosis in response to many cellular stresses, and it can be activated through post-translational modifications, such as phosphorylation, acetylation, and ubiquitination [20,21]. The inactivation or deletion of the P53 gene is one of the distinct features seen in the early, as well as the late stages of GBM. It has been reported that the mutations of p53 participate in disease progression, especially from low-grade astrocytoma (grade II) or anaplastic astrocytoma (grade III) [22,23].

Furthermore, the activation of the canonical Wnt pathway, which is controlled by the adenomatous polyposis coli (APC) protein, has been identified as the driving force behind several human cancers, including colorectal cancer and GBM [24,25]. The degradation of β-catenin by APC prevents the activation of Wnt targeted genes that stimulate cell division and thereby prevents cell overgrowth [26,27]. Additionally, the APC protein plays a crucial role in ensuring the correct chromosome number in the cell nucleus after division [8]. During early brain development, Wnt signaling plays a critical role in mediating brain patterning and maturation, both before and after birth [28,29]. The involvement of Wnt signaling in brain tumors was initially discovered through the observation of germline mutations of APC in patients with malignant glioma and medulloblastoma [30,31,32]. In recent years, research has revealed that certain transcription factors, such as ASCL1, have a direct role in activating Wnt signaling. ASCL1 is essential for maintaining glioblastoma multiforme cancer stem cells and promoting in vivo tumorigenicity [33,34]. Meanwhile, other studies have reported that the inactivation of certain Wnt pathway inhibitor genes, leading to the activation of Wnt signaling pathways, is a common event during astrocytic glioma carcinogenesis and is associated with malignant progression [14,34,35]. Additionally, increased β-catenin expression has consistently been found in the late stages of astrocytic tumors and has been correlated with poor prognosis and shorter survival in glioblastoma multiforme patients [36,37]. Furthermore, EGFR expression is upregulated in primary glioblastoma multiforme, and it correlates with tumor malignancy [38]. EGFR signaling through extracellular signal-regulated kinases 1/2 and casein kinase-2 in glioma cells leads to the phosphorylation of α-catenin and promotes β-catenin transactivation, which is associated with a high grade of glioma malignancy [39,40]. Consequently, the Wnt/β-catenin pathway may play an important role not only in brain development but also in the progression of GBM [41].

In this study, we utilized a GFAP Cre transgenic strain to investigate the role of APC haploinsufficiency in conjunction with mutant Kras^G12D^ activation and p53 loss, leading to the induction of GBM tumorigenesis. The GFAP-Cre; Kras^G12D^; APC^L/+^; P53^L/L^ mouse model demonstrates an increased activation of the WNT/β-catenin pathway and exhibits cancer-stem-cell-like properties, closely resembling human glioblastoma multiforme’s histopathological features. As a result, our GBM mouse model presents a valuable in vivo tool for studying the pathogenesis of GBM and will create opportunities to discover novel biomarkers for early GBM diagnosis and the treatment of anaplastic gliomas and GBM.

## 2. Materials and Methods

### 2.1. Generation of GFAP-Cre/Kras^G12D^/P53^L/L^ (GKP) and GFAP-Cre/Kras^G12D^/APC^L/+^/P53^L/L^ (GKAP) Mice

GFAP-Cre mice (purchased from Jackson Laboratory), LSL-Kras^G12D^ mice (originally described by T. Jacks et al.), and P53 conditional knockout floxp mice (originally described by A. Berns et al.) were maintained as previously documented [42,43]. APC mice were obtained from the NCI Mouse Repository (Strain: B6 Cg-Apctm2Rak; Strain number 01XAA) [43,44]. The genetic manipulation in these mice targeted oncogenic Kras (K-ras^G12D^) to the endogenous KRAS locus, with the K-ras allele featuring a stop codon flanked by loxP sites, and the conditional APC allele having two loxP sites flanking exon 14. The Cre-recombinase-mediated excision of the stop cassette facilitated the expression of oncogenic Kras from the endogenous Kras promoter. GFAP-Cre transgenic mice (referred to as GFAP-Cre mice) were crossed with Kras^G12D^ mice to generate GFAP-Cre/Kras^G12D^ mice, with Cre-recombinase expression driven by the GFAP promoter. Subsequently, GFAP-Cre/Kras^G12D^ mice were bred with P53^L/L^ mice to produce GFAP-Cre/Kras^G12D^/P53^L/L^ (GKP) mice, and GFAP-Cre/P53^L/L^ mice were further crossed with Kras^G12D^/APC^L/L^/P53^L/L^ mice to create GFAP-Cre/Kras^G12D^/APC^L/+^/P53^L/L^ (GKAP) compound mice.

### 2.2. Genotyping of Mutant Compound Mice

The mouse genotyping protocols were based on recommendations from Jackson Laboratories (Bar Harbor, ME, USA) [43,44]. In brief, we used 2 μL DNA, 0.4 μL 2.5 mM dNTPs, 0.4 μL Taq DNA polymerase, 2 μL 10X reaction buffer and 0.5 μL primer. Reaction protocol is as follows: 94 °C for 3 min, followed by 35 cycles of 94 °C, 64 °C, and 72 °C for 30 s at each temperature, then 72 °C for 3 min, and holding at 4 °C. Genotyping primers for the detection of GFAP-Cre, Kras^G12D^, Apc^L/L^ and p53^L/L^ genetically modified alleles are listed in Appendix A.

### 2.3. Behavioral Assessment of Transgenic Mice

The mice were regularly observed at least 2 to 3 times a week. Behavioral tests for mutant mice were categorized into several types, including frailty, lameness (crippling), inclination walking (tilting), wiggling (teetering), seizures (epilepsy), and headstand circling. Mice were sacrificed either at the end of the observation period or if they experienced a weight loss of 20% to 30%. The observed behaviors of the mice were recorded before autopsy and histological analysis.

### 2.4. Histology Analysis

Mice were euthanized humanely with carbon dioxide, followed by the fixation of their brains in 10% buffered formalin acetate and subsequent fixation in 75% ethanol. The brains were divided into hemispheres, with one hemisphere subjected to parasagittal sections. Tissue sections were then embedded in paraffin and then sectioned at 5 µm thickness and were stained with hematoxylin and eosin (H&E) for histological examination [45].

### 2.5. Immunohistochemistry (IHC) and Immunofluorescence (IF) Analysis

Tumor samples were processed for immunohistochemistry (IHC) by isolating fixed tumors, embedding them in paraffin, and sectioning them into 5 µm thick slices. These paraffin-embedded sections were mounted on glass slides and dried at 60~65 °C for 30 min to let the paraffin wax melt. Deparaffinization was further carried out in xylene for 1 h, followed by washing with a graded series of ethanol (ranging from 95% to 50%). The sections underwent antigen unmasking via boiling in a high-pH antigen unmasking solution (VECTOR^®^ Laboratories, Newark, CA, USA, High pH, Cat. No. H-3301) for 5 min. Endogenous peroxidase activity was blocked with 0.3~3% H_2_O_2_ for 5 min, followed by a 10 min rinse in distilled water. Non-specific binding was prevented using blocking agents such as goat serum, horse serum, or mouse Ig blocking reagent (Vector^®^ M.O.M.™). Primary antibodies were applied and allowed to incubate overnight at 4 °C. Following washes with PBST and PBS, secondary antibodies were applied for 1 h. To visualize the staining, Avidin/Biotinylated enzyme Complex was used, and the DAB (3, 3′-diaminobenzidine) HRP substrate produced a dark-brown reaction (Vector Laboratories). Finally, the sections were counterstained with hematoxylin and covered with a glass cover slip. Various primary antibodies employed in these IHC analyses are listed in Appendix A. In addition, immunofluorescence staining (IF) was performed, as described previously [45,46].

### 2.6. TUNEL Assay

The fluorometric TUNEL (terminal deoxynucleotidyl transferase dUTP nick-end labeling) assay was carried out following the manufacturer’s protocol as specified in the Dead End Fluorometric TUNEL system from Promega (Madison, WI, USA). This assay is designed to detect apoptotic cells by marking fragmented DNA with fluorescein-12–dUTP. The analysis was performed following the procedures described previously [47].

### 2.7. Cell Culture

Primary GBM cell lines were isolated from brain tumors from GFAP-Cre/Kras^G12D^/P53^L/L^ and GFAP-Cre/Kras^G12D^/APC^L/+^/P53^L/L^ transgenic mice. Tissue was transferred to fresh PBS, washed once, and then mechanically dissociated using trypsin to digest tissues to cell lines in a 37 °C water bath for 10 min. All cell lines were grown in Roswell Park Memorial Institute medium (RPMI, Buffalo, NY, USA) and supplemented with 10% fetal bovine serum and 1% penicillin/streptomycin and were grown in a humidified incubator containing 5% CO_2_ at 37 °C.

### 2.8. Cell Viability Assay

In a 96-well plate, 2 × 10^4^ cells were initially seeded and incubated overnight. The following day, 2% fetal bovine serum was introduced, and the cells were cultured for 5 days. After this incubation period, each well received 20 μL of 5 mg/mL MTT (thiazolyl blue tetrazolium bromide) from PROTECH (Sparks, NV, USA), along with 200 μL of medium, and were then incubated for an additional 2 h to facilitate the reaction. Subsequently, the medium was aspirated, and the crystals formed were completely dissolved using 200 μL of DMSO (Sigma Aldrich, Saint Louis, MO, USA). The optical density (OD) at 570 nm was determined using a Bio-Rad (Hercules, CA, USA) iMark™ microplate absorbance ELISA reader.

### 2.9. Western Blotting

Cell pellets or brain tissue lysates were collected and lysed using 100 µL of protein lysis buffer (containing 2 mmol/L EDTA, 150 mmol/L NaCl, 0.5% NP-40, 0.1% SDS) for 2 min on ice. The protein concentration was determined using the BCA Protein Assay Reagent (Pierce, Appleton, WI, USA). Subsequently, 30 μg of total protein was separated on polyacrylamide gels with varying percentages and transferred onto polyvinylidene difluoride (PVDF) membranes. These PVDF membranes were blocked with 4% non-fat dry milk in Tris-buffered saline and Tween 20 (TBST), followed by an overnight incubation with primary antibodies at 4 °C. The following day, secondary antibodies, such as peroxidase-conjugated anti-rabbit (1:2000, Santa Cruz, Dallas, TX, USA) and anti-mouse (1:2000, Santa Cruz), were applied for 1 h at room temperature. Protein detection was performed using the Immobilon Western Chemiluminescent HRP substrate kit (Millipore, Waltham, MA, USA). Information for the antibodies used for Western blotting is listed in Appendix A. Subsequently, the intensity of the bands was further measured using the software ImageJ 1.45s (National Institutes of Health—NIH, Bethesda, MD, USA; https://imagej.nih.gov/ij/). Fold change relative to control was calculated using indicated protein/GAPDH ratio.

### 2.10. Statistical Analysis

The results are expressed as the means ± S.E. from at least three independent experiments. The statistical significances of differences between groups were determined using Student’s *t*-test (* *p* < 0.05; ** *p* < 0.01).

## 3. Results

### 3.1. Generation and Characterization of GFAP-Cre; Kras^G12D^; APC^L/+^; p53^L/L^ Mice

In a recent study, researchers delved into the molecular mechanisms underpinning glioma tumorigenesis using genetically modified mice. They employed glial-fibrillary-acidic-protein (GFAP)-driven Cre glioblastoma mouse models, the allowing precise recombination of the activation of the Kras^G12D^ oncogene and the homozygous deletion of the p53 tumor suppressor gene specifically within astrocyte lineage cells. Previous studies had demonstrated that this combination of genetic alterations led to the development of high-grade gliomas [42]. However, the crucial question of whether these mice would progress to GBM remained unexplored. Meanwhile, the canonical Wnt signaling pathway has been linked to glioma formation, with the aberrant activation of Wnt signaling often attributed to the inactivation of the APC gene, seen in various cancers, including colorectal cancer [48]. Thus, to investigate the impact of APC inactivation on GBM tumorigenesis, we employed a genetically engineered mouse with APC floxp conditional alleles and crossed it with models featuring p53 loss and Kras^G12D^ activation, all facilitated by the GFAP-Cre transgene. This intricate genetic design yielded two distinct mouse models: GFAP-Cre; Kras^G12D^; p53^L/L^ and GFAP-Cre; Kras^G12D^; APC^L/+^; p53^L/L^ (Figure 1A). In the latter model, the APC gene underwent selective inactivation through the Cre-mediated deletion of exon 14, ultimately resulting in a reduction in APC protein expression during glioma tumorigenesis.

Subsequently, to confirm the genotypes of these compound mice, specific genotyping PCR analysis was performed using mouse tail DNA samples (Figure 1B). In these aging compound mouse colonies, a gradual decrease in body weight was observed in both GFAP-Cre; Kras^G12D^; p53^L/L^ and GFAP-Cre; Kras^G12D^; APC^L/+^; p53^L/L^ mice compared to the control mice (GFAP-Cre; P53^L/L^) (Figure 1C). Furthermore, the results revealed that the genetic alterations induced in these compound mice led to tumorigenic consequences and impacted their overall survival time. Of note, GFAP-Cre; Kras^G12D^; APC^L/+^; p53^L/L^ mice displayed a median survival time of 55 days, while GFAP-Cre; Kras^G12D^; p53^L/L^ mice presented a slightly longer median survival time of around 65 days (Figure 1D). These survival durations were significantly shorter than those of the control mice (GFAP-Cre; P53^L/L^) or wild-type mice. Our results suggest that the inactivation of APC, in combination with Kras^G12D^ activation and p53 loss, expedites glioblastoma development and may contribute to the development of GBM.

### 3.2. Conditional APC Haploinsufficiency Combined with Mutant Kras Activation and p53 Loss Cooperatively Induces the Development of GBM in Mice

To examine whether APC deletion in GFAP-Cre; Kras^G12D^; p53^L/L^ compound mice drives GBM formation in vivo, we usually compare phenotypic differences in gross appearance, behavior, and pathohistology between GFAP-Cre; P53^L/L^, GFAP-Cre; Kras^G12D^; p53^L/L^, and GFAP-Cre; Kras^G12D^; APC^L/+^; p53^L/L^ mice. Remarkably, most GFAP-Cre; APC^L/L^ mice experience early postnatal lethality due to congenital hydrocephalus. Notably, GFAP-Cre; Kras^G12D^; APC^L/+^; p53^L/L^ mice display diminished activity, accompanied by head imbalance, and exhibit a significantly enlarged and deformed skull at a very young age (Figure 2A and Appendix A). The histological examination of brain tumor sections from these mice reveals several hallmark features of human glioblastoma (GBM), including multinucleated giant cells, abundant eosinophilic cytoplasm, and large vesicular nuclei with prominent nucleoli. Hematoxylin and eosin (H&E) staining highlights key aspects of human glioma, such as hemorrhage, increased cellularity, nuclear vascular proliferation, and necrosis in the brain tumor tissues of GFAP-Cre; Kras^G12D^; APC^L/+^; p53^L/L^ mice (Figure 2Bi–Biii). Our quantitative analysis confirmed significant differences in CD31+ microvessel density and giant cell count between the GFAP-Cre; P53^L/L^, GFAP-Cre; Kras^G12D^; p53^L/L^ and GFAP-Cre; Kras^G12D^; APC^L/+^; p53^L/L^ groups (Figure 2Biii). Conversely, the examination of the brains of GFAP-Cre; Kras^G12D^; p53^L/L^ mice reveals a few high-grade glioma characteristics, including heterogeneous lesions, spindle cells, large and elongated nuclei, bipolar processes, and a few giant cells. However, these mice lack features such as vascular proliferation or necrosis within their brain lesions (Figure 2Bi–Biii). To further explore the potential implications of these findings in human glioblastoma, we established primary GBM cell lines wherein cell lines were maintained in RPMI 10% FBS-containing medium and these primary murine GBM cells grew as monolayers. All of these murine primary cell lines displayed a malignant glial-like phenotype. Their identification as glioblastoma cell lines was solidified by confirming their immunopositivity for the glial marker GFAP (Figure 2C). This confirmation opens up opportunities for the further exploration and analysis of molecular mechanisms in our upcoming experiments.

### 3.3. Coordination between Cell Proliferation and Apoptosis in GBM with APC Haploinsufficiency

To investigate the molecular mechanism underlying the in vivo effects of APC haploinsufficiency-enhanced GBM progression, we examined cell death (TUNEL assay) and cell proliferative (Ki-67 and Phospho-Histone 3 markers) activities using immunohistochemistry (Figure 3Ai,Aii). The TUNEL assay, a widely used method to detect apoptotic cells, was employed to explore cell death in the context of APC haploinsufficiency in GBM. The findings from this assay were remarkable, indicating that brain lesions derived from GFAP-Cre; Kras^G12D^; APC^L/+^; p53^L/L^ mice exhibited a higher prevalence of necrotic areas. Necrosis, characterized by cell death and tissue breakdown, is a hallmark of aggressive malignant behavior. This observation underscores the role of APC haploinsufficiency in promoting GBM’s invasive and destructive nature.

In addition to examining cell death, the study further investigated cell proliferation by utilizing markers such as Ki-67 and Phospho-Histone 3 (pHistone3). Ki-67 is a protein associated with cell division and is frequently used as a marker of proliferative activity, while Phospho-Histone 3 is a marker of cells undergoing mitosis, another key process in cell proliferation. The results of our immunohistochemical analysis revealed that brain lesions in GFAP-Cre; Kras^G12D^; APC^L/+^; p53^L/L^ mice exhibited significantly higher levels of Ki-67 and Phospho-Histone 3 staining compared to normal brain tissues. This heightened proliferation activity suggests that APC haploinsufficiency contributes to the rapid and uncontrolled growth of GBM tumors. Moreover, to further investigate the impact of APC haploinsufficiency on cell proliferation, the study conducted MTT assays to compare the proliferation rates of GFAP-Cre; Kras^G12D^; APC^L/+^; p53^L/L^ cell lines with those of GFAP-Cre; Kras^G12D^; p53^L/L^ cell lines in an in vitro setting. Strikingly, our in vitro results indicated that there were no significant differences in the proliferation rates between these two cell lines (Figure 3B). This observation suggests that the effect of APC haploinsufficiency on cell proliferation may be more prominent in the in vivo tumor microenvironment. One intriguing finding that emerged from our study was the increased tumorigenic ability of GFAP-Cre; Kras^G12D^; APC^L/+^; p53^L/L^ cells when compared to GFAP-Cre; Kras^G12D^; p53^L/L^ cells, as demonstrated in in vitro colony formation assays (Figure 3C). Tumorigenicity refers to the capacity of a cell or cell population to form tumors, and our result suggests that APC haploinsufficiency has a more pronounced effect on the tumorigenic potential of GBM cells, influencing their ability to grow and form tumor-like structures, which suggests that APC haploinsufficiency not only influences the rate of cell proliferation but also plays a significant role in promoting the tumorigenicity of GBM cells.

### 3.4. APC Haploinsufficiency in GBM Leads to Aberrant Activation of the WNT/β-Catenin Signaling Pathway

The APC protein is a negative regulator that controls β-catenin activity, and β-catenin serves as a dual-function protein, regulating the coordination of cell–cell adhesion and Wnt-pathway-related gene transcription [49]. Therefore, a lack of APC allele or an abundance of it will modulate the impacts of the Wnt pathway signaling responses. We have shown that GFAP-Cre; Kras^G12D^; APC^L/+^; p53^L/L^ has a high expression of β-catenin, c-myc, and cyclin D1 via immunostaining (Figure 4A). We detected the expression levels of β-catenin, c-myc, and cyclin D1 proteins, which are important downstream molecules in the Wnt signaling pathway, in the tumors derived from the brain lesions of GFAP-Cre; Kras^G12D^; APC^L/+^; p53^L/L^ mice. Our data demonstrated that the expression levels of β-catenin, c-myc, and cyclin D1 protein were significantly increased in GFAP-Cre; Kras^G12D^; APC^L/+^; p53^L/L^ mouse tissues compared to the brains tissues derived from GFAP-Cre; Kras^G12D^; p53^L/L^ mice. We further examined their protein expression levels from primary GBM tumor cell lines. Western blot analysis also confirmed that β-catenin, c-Myc, and cyclin D1 proteins are highly expressed in GFAP-Cre; Kras^G12D^; APC^L/+^; p53^L/L^ cells compared to GFAP-Cre; Kras^G12D^; p53^L/L^ cells (Figure 4B), Our data suggested that the inactivation of APC in GBM may lead to the aberrant activation of the WNT/β-catenin signaling pathway to promote GBM progression.

### 3.5. APC Haploinsufficiency Amplified Cancer Stemness in GBM Mice

Subsequently, we conducted an immunohistochemistry analysis to investigate the expression of vimentin, a well-known astrocytic marker associated with glial cells. Our analysis unveiled the presence of positive foci within these GBM tissues, notably concentrated in areas displaying astrocytic characteristics. Significantly, these positive foci were more pronounced in the brain sections of GFAP-Cre; Kras^G12D^; APC^L/+^; P53^L/L^ group compared to the control group. Next, we further explored the expression of the cancer stem cell markers Notch1, Nestin, and CD133 in GBM. Our findings revealed elevated levels of Notch1, CD133, and Nestin expression in glioblastomas from the GFAP-Cre; Kras^G12D^; APC^L/+^; p53^L/L^ mice when compared to brain sections from the GFAP-Cre;Kras^G12D^;p53^L/L^ group. Immunostaining and Western blot analysis underscored the pivotal role of CD133 and Nestin in the GFAP-Cre; Kras^G12D^; APC^L/+^; p53^L/L^ GBM mouse model (Figure 5A,B).

Meanwhile, PDGFRα amplifications are found in approximately 13% of GBMs and are particularly enriched in proneural subtype tumors [50]. Interestingly, PDGF-A has been shown to have an impact on the fate of stem cells residing in the subventricular zone (SVZ) by promoting the differentiation of oligodendrocyte precursor cells (OPCs). Histologically, in GFAP-Cre PDGFA transgenic mice, spontaneous gliomas exhibited characteristics of both astrocytes and oligodendrocytes and were classified as WHO grade III oligoastrocytomas [42]. Moreover, PDGFR-α expression has been reported to be elevated in both low- and high-grade astrocytomas, indicating its involvement in tumor cell proliferation at various stages of glioma development [51]. This suggests a substantial role for PDGFR-α in regulating cancer stemness in glioblastoma GBM. In our study, we also confirmed higher levels of PDGFR-α protein expression in GBMs derived from GFAP-Cre; Kras^G12D^; APC^L/+^; p53^L/L^ and GFAP-Cre; Kras^G12D^; p53^L/L^ mice in comparison to the GFAP-Cre; Kras^G12D^; p53^L/L^ group (Figure 5A,B).

### 3.6. APC Haploinsufficiency Leads to Increased Angiogenesis and Activates the EGFR Pathway in GBM

A prominent characteristic of human glioblastoma (GBM) is the presence of angiogenic blood vessels, indicating a high degree of vascularization. To gain a deeper understanding of the microenvironment and the signals driving proliferation and angiogenesis, we conducted an investigation into the angiogenic properties of the brains of GFAP-Cre; Kras^G12D^; APC^L/+^; p53^L/L^ mice. Our findings also revealed that key signaling pathways, including epidermal growth factor receptor (EGFR) and vascular endothelial growth factor (VEGF), were activated in these brain tissues (Figure 6A). This activation suggests a strong pro-angiogenic environment, consistent with the angiogenic character of human GBM.

Next, we further explored the downstream signaling pathways mediated by EGFR, specifically the PI3K/AKT and MAPK/ERK pathways. Our examination revealed elevated levels of phosphorylated AKT and ERK in the GFAP-Cre; Kras^G12D^; APC^L/+^; p53^L/L^ group, indicating increased activity in these kinase signaling cascades. Western blot analysis demonstrated a decrease in EGFR protein levels in the GFAP-Cre; Kras^G12D^; p53^L/L^ group compared to GFAP-Cre; Kras^G12D^; APC^L/+^; p53^L/L^ group (Figure 6B). These results collectively confirm that APC haploinsufficiency, in conjunction with the activation of WNT/β-catenin and EGFR signaling pathways, drives both cell proliferation and angiogenesis, ultimately leading to the formation of GBM. Our study provides valuable insights into the molecular processes underlying GBM progression, particularly the role of APC haploinsufficiency in activating critical signaling pathways. The increased angiogenic character (VEGF^high^) observed in the GFAP-Cre; Kras^G12D^; APC^L/+^; p53^L/L^ mice suggests that these pathways contribute to the vascularization of brain tumors, mirroring a key feature of human GBM.

## 4. Discussion

Numerous studies in the field of glioblastoma research have provided compelling evidence of the genetic alterations that drive the development of this aggressive form of brain cancer. One recurring theme in these studies concerns the frequent deletions and mutations observed in the PTEN lipid phosphatase tumor suppressor genes. The PTEN gene, short for phosphatase and tensin homolog, acts as a tumor suppressor by regulating cell growth, division, and survival. PTEN achieves this by dephosphorylating a critical signaling molecule known as phosphatidylinositol-3,4,5-trisphosphate (PIP3), which plays a pivotal role in the RTK/PI3K pathway. The RTK/PI3K pathway, comprised of receptor tyrosine kinases (RTKs) and phosphoinositide 3-kinases (PI3Ks), is responsible for orchestrating various cellular processes, including proliferation, differentiation, and survival [52]. Remarkably, genetic anomalies in PTEN have been detected in approximately 86% of glioblastoma samples, shedding light on their significant role in the pathogenesis of this devastating disease [44,53].

In glioblastoma, the intricate network of genetic changes involves not only PTEN but also other key players within the RTK/PI3K pathway. Among these, Kras stands out as a gene that, when mutated, can exert similar downstream effects to PTEN alterations. These effects include uncontrolled cell growth and division, hallmarks of tumor development [54]. Therefore, the co-occurrence of PTEN mutations and genetic events in the RTK/PI3K pathway, including Kras mutations, is suggested to have similar effects in promoting glioblastoma tumorigenesis. Another critical facet of gliomagenesis involves the inactivation of the p53 tumor suppressor pathway, occurring late during the disease’s progression. The p53 protein, often referred to as the “guardian of the genome”, plays a pivotal role in maintaining genomic stability. It does so by monitoring the integrity of the DNA and initiating protective mechanisms, such as cell cycle arrest and DNA repair, in response to genetic damage [55]. In cases where the damage is beyond repair, p53 can trigger apoptosis, a form of programmed cell death. Many studies have demonstrated that the inactivation of p53 during gliomagenesis is associated with the emergence of certain high-grade gliomas and GBM [10,56]. This underscores the significance of the p53 pathway in preventing the development of glioblastoma by safeguarding the genome against mutations and genomic instability.

To gain deeper insights into the mechanisms driving brain carcinogenesis and the development of GBM, researchers have turned to genetically modified disease mouse models. One such model, the GFAP-Cre; Kras^G12D^; p53^L/L^ mouse model, has been engineered to mimic the genetic alterations observed in human glioblastoma. This transgenic mouse model involves the activation of the Kras^G12D^ oncogene and the deletion of p53 tumor suppressor genes, effectively mirroring the genetic events that contribute to gliomagenesis [56]. Notably, recent studies have reported that this model exhibits high-grade glioma phenotypes, validating its utility as a research tool for investigating glioblastoma development [42,56]. Meanwhile, while the PTEN, RTK/PI3K, and p53 pathways have provided valuable insights into glioblastoma, there is emerging evidence that the canonical Wnt pathway also plays a pivotal role in driving glioma tumorigenesis [57]. The canonical Wnt pathway is a highly conserved signaling cascade involved in critical cellular processes, including embryonic development, tissue homeostasis, and cancer progression [58]. The dysregulation of this pathway can lead to uncontrolled cell growth and division, making it a compelling candidate for further investigation in the context of glioblastoma. In light of this, we therefore expanded the repertoire of disease models to include the GFAP-Cre; KrasG12D; APC^L/+^; p53^L/L^ mouse model. This sophisticated model integrates the activation of the Kras^G12D^ oncogene, the deletion of p53 tumor suppressor genes, and the inclusion of the APC^L/+^ alteration to study the canonical Wnt pathway’s activation and its role in the regulation of glioblastoma tumorigenesis. By combining these genetic alterations, we further dissected the intricate crosstalk between these pathways, shedding light on the complex molecular mechanisms driving the development of GBM.

First, we observed that GFAP-Cre; Kras^G12D^; APC^L/+^; p53^L/L^ mice exhibited wasting and weakness accompanied by poor head balance and presented with an extremely enlarged and deformed skull at around 6–8 weeks old. Histopathologic heterogeneity in glioblastoma is a hallmark during GBM progression. The GFAP-Cre; Kras^G12D^; APC^L/+^; p53^L/L^ mice displayed many significant features of human gliomas, including more giant cell formation, hemorrhage, increased cellularity, vascular proliferation, and necrosis, which lead to classification as a grade IV glioma. We further established GBM primary cell lines from an APC haploinsufficiency GBM mouse model. This provides an opportunity to explore the molecular pathogenic mechanism for the development of GBM in our compound mice. We confirmed that APC haploinsufficiency led to the increased expression of the downstream molecules in the Wnt signaling pathway, such as β-catenin, c-myc and cyclin D1 protein. Our data also demonstrated that the increased in vivo tumor cell proliferation, cancer stemness and angiogenesis occurred via the activation of the WNT/β-catenin signaling pathway during GBM carcinogenesis.

Next, to shed light on the molecular mechanisms underlying glioblastoma development in the GFAP-Cre; Kras^G12D^; APC^L/+^; p53^L/L^ mouse model, our study examined the expression of various markers. We demonstrated that one of the glial cell astrocytic markers, vimentin, was positive in tumor tissue, particularly in areas displaying astrocytic appearances. Furthermore, the cancer stemness phenotype plays a pivotal role in tumor formation, often resulting in tumors with a lower degree of differentiation due to their self-renewing capacity and plasticity [59,60]. In the context of glioma, several studies have highlighted the significance of glioma stem cell markers, specifically CD133 and Nestin, in determining the prognosis of gliomas patients [60,61]. A high expression of CD133 has been identified as an independent risk factor for poor prognosis in glioblastoma or GBM, especially in cases classified as WHO grade IV gliomas [62]. On the other hand, high Nestin expression has been associated with the prognosis of patients with WHO grade II–III gliomas [63,64,65]. Additionally, the platelet-derived growth factor receptor (PDGFR), a major mitogen receptor for connective tissue cells and glia, has been implicated in high-grade astrocytomas. This suggests that the PDGFRα or PDGFRβ pathway plays a role in cancer cell proliferation during the various stages of glioma development [66,67]. Furthermore, the intricate network of signaling pathways within cancer cells allows the integration of a multitude of external and internal stimuli, resulting in uncontrolled proliferation. Thus, the study also demonstrated the activation of the EGFR signaling pathway, known for its role in regulating the proliferation and migration of neural stem cells, in GFAP-Cre; Kras^G12D^; APC^L/+^; p53^L/L^ GBM mice. It was hypothesized that active EGFR-mediated downstream signaling pathways, such as PI3K/AKT and MAPK/ERK, may lead to an increase in VEGF protein levels in GFAP-Cre; Kras^G12D^; APC^L/+^; p53^L/L^ mice. Notably, the epidermal growth factor receptor (EGFR) and the Wnt/β-catenin pathways have been also shown to interact and synergize during glioma tumorigenesis [68]. Our data suggested that APC haploinsufficiency may enhance EGFR and VEGF signaling pathways, contributing to cell proliferation and angiogenesis and, ultimately, driving the progression to GBM.

In summary, this study presents compelling evidence that the GFAP-Cre; Kras^G12D^; APC^L/+^; p53^L/L^ mouse model can promote GBM tumorigenesis, particularly when combined with the activation of the WNT/β-catenin signaling pathway. The study highlights the importance of CD133, a cancer stem cell marker, and the EGFR and VEGF signaling pathways in this GBM mouse model. Consequently, this model offers a valuable in vivo platform for future research aimed at discovering novel biomarkers for early diagnosis and conducting preclinical studies to evaluate potential new agents for targeted GBM therapy. This research has the potential to significantly advance our understanding of glioblastoma and improve the therapeutic strategies for this challenging disease.

## 5. Conclusions

Our study highlights the multifaceted genetic landscape underlying the development of glioblastoma (GBM) and the pivotal role of various signaling pathways in driving this aggressive form of brain cancer. Here, we expanded our understanding by incorporating the canonical Wnt pathway through the GFAP-Cre; Kras^G12D^; APC^L/+^; p53^L/L^ mouse model, shedding light on the intricate crosstalk between these pathways. Our findings underscore the significance of cancer stemness markers such as CD133, as well as the role of PDGFR-α in the progression of glioma towards GBM. Additionally, our study provides insights into the activation of the EGFR and VEGF signaling pathways, contributing to increased cell proliferation and angiogenesis. The GFAP-Cre; Kras^G12D^; APC^L/+^; p53^L/L^ mouse model proves to be a valuable in vivo platform for further research, offering potential biomarkers for early diagnosis and serving as a preclinical tool for the development of innovative GBM therapies.

## Figures and Tables

**Figure 1 cancers-16-01046-f001:**
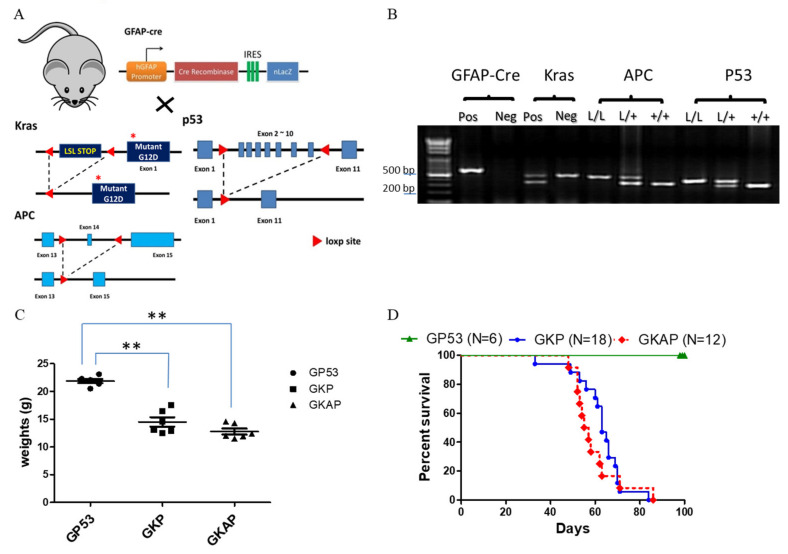
Targeting mutant Kras activation with APC and p53 deletion to induce GBM formation in mice. (**A**) Schematic diagram for the genetically modified GBM mouse mating setup. (**B**) Specifically genotyping PCR was conducted for the genotyping of animals and the detection of the LoxP product of each targeted allele, with PCR products visualized via agarose gel electrophoresis and displaying the expected sizes: GFAP-Cre (600 bp), Kras size (LoxP: 327 bp; wild type: 450 bp), APC (LoxP: 430 bp; wild type: 320 bp) and P53 (LoxP: 370 bp; wildtype: 288 bp). Weight curve (**C**) and survival (**D**) alterations for GFAP-Cre; p53^L/L^ (GP53 serves the normal control group), GFAP-Cre; Kras; p53^L/L^ (GKP) and GFAP-Cre; Kras; APC^L/+^; p53^L/L^ (GKAP) mice. ** *p* < 0.01.

**Figure 2 cancers-16-01046-f002:**
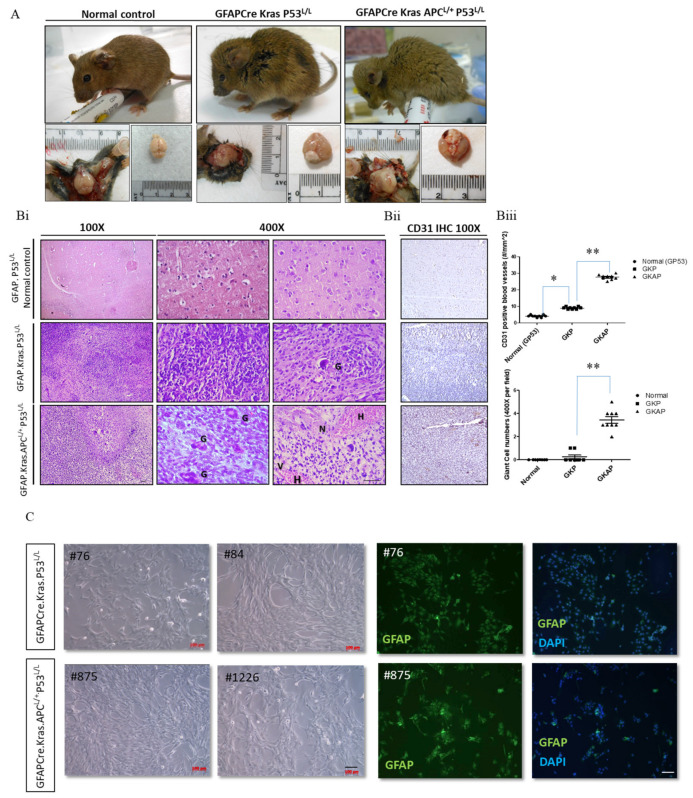
Pathohistological analyses showing glioblastoma formation in GFAP-Cre; Kras^G12D^; P53^L/L^ and GFAP-Cre; Kras^G12D^; APC^L/+^; P53^L/L^ mice. (**A**) Gross anatomy analysis of GFAP-Cre; P53^L/L^ (GP53 normal control), GFAP-Cre; Kras^G12D^; P53^L/L^ (GKP), and GFAP-Cre; Kras^G12D^; APC^L/+^; P53^L/L^ (GKAP) mice. (**B**) (**i**) Representative hematoxylin-and-eosin (H&E)-stained brain sections (the temporal lobes) representing different indicated genotypes; GFAP-Cre; P53^L/L^ control; GFAP-Cre; Kras^G12D^; P53^L/L^ and GFAP-Cre; Kras^G12D^; APC^L/+^; P53^L/L^ mice. Key features include multinucleated giant cells (G), hemorrhage (H), increased cellularity, vascularity (V), and necrosis (N). Bars: 100 μm. (**ii**) IHC staining of CD31 on the brain sections derived from GFAP-Cre; P53^L/L^, GFAP-Cre; Kras^G12D^; P53^L/L^, and GFAP-Cre; Kras^G12D^; APC^L/+^; P53^L/L^ mice. (**iii**) The numbers of CD31-labeled blood vessels and giant cells were counted in six non-overlapping microscopic fields of the indicated genotypes. * *p* < 0.05; ** *p* < 0.01. (**C**) Primary cell cultures derived from the brains of GFAP-Cre; Kras^G12D^; P53^L/L^, and GFAP-Cre; Kras^G12D^; APC^L/+^; P53^L/L^ mice. Morphological characteristics of GFAP-Cre; Kras^G12D^; P53^L/L^ (#76; #84), and GFAP-Cre; Kras^G12D^; APC^L/+^; P53^L/L^ (#875; #1226) cells are presented, respectively. Murine GBM primary cell lines, confirmed as glioblastoma, displayed GFAP glial marker immunopositivity (FITC) with DAPI-stained blue nuclei. Bars: 100 μm.

**Figure 3 cancers-16-01046-f003:**
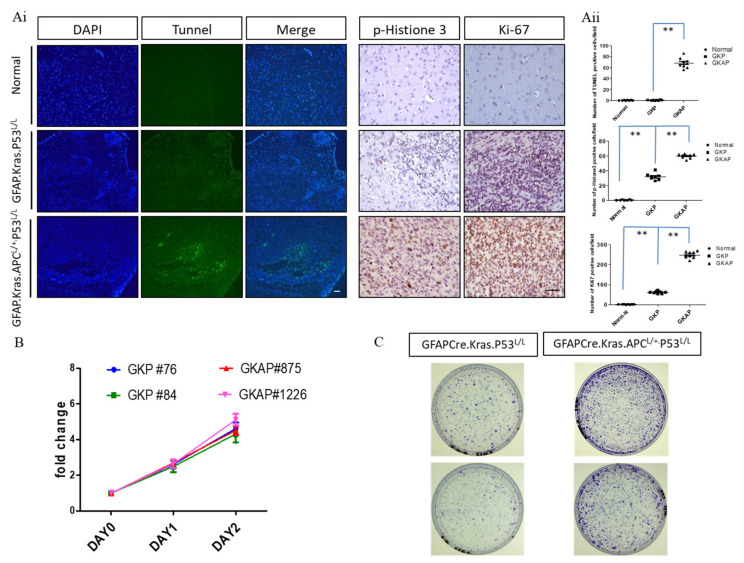
Analysis of proliferative and apoptotic activities in murine GBM. (**Ai**) TUNEL assays were conducted to compare the extent of necrosis in GFAP-Cre; Kras^G12D^; APC^L/+^; P53^L/L^ (GKAP) brain sections with the indicated genotypes. Cell proliferation was evaluated through whole-mount immunohistochemistry using anti-phosphohistone H3 and anti-Ki-67 antibodies, as indicated. Bars: 100 μm. (**Aii**) The percentage of TUNEL-, Ki67-, and pHistone3-positive cells were quantified by ImageJ software ver. 1.45. ** *p* < 0.01. (**B**) Proliferative rates were determined via MTT assays on primary cells cultured from the brains of GFAP-Cre; Kras^G12D^; P53^L/L^ and GFAP-Cre; Kras^G12D^; APC^L/+^; P53^L/L^ mice. (**C**) The colony formation assays demonstrated the impact of APC haploinsufficiency on the ability of murine primary GBM cells to form colonies.

**Figure 4 cancers-16-01046-f004:**
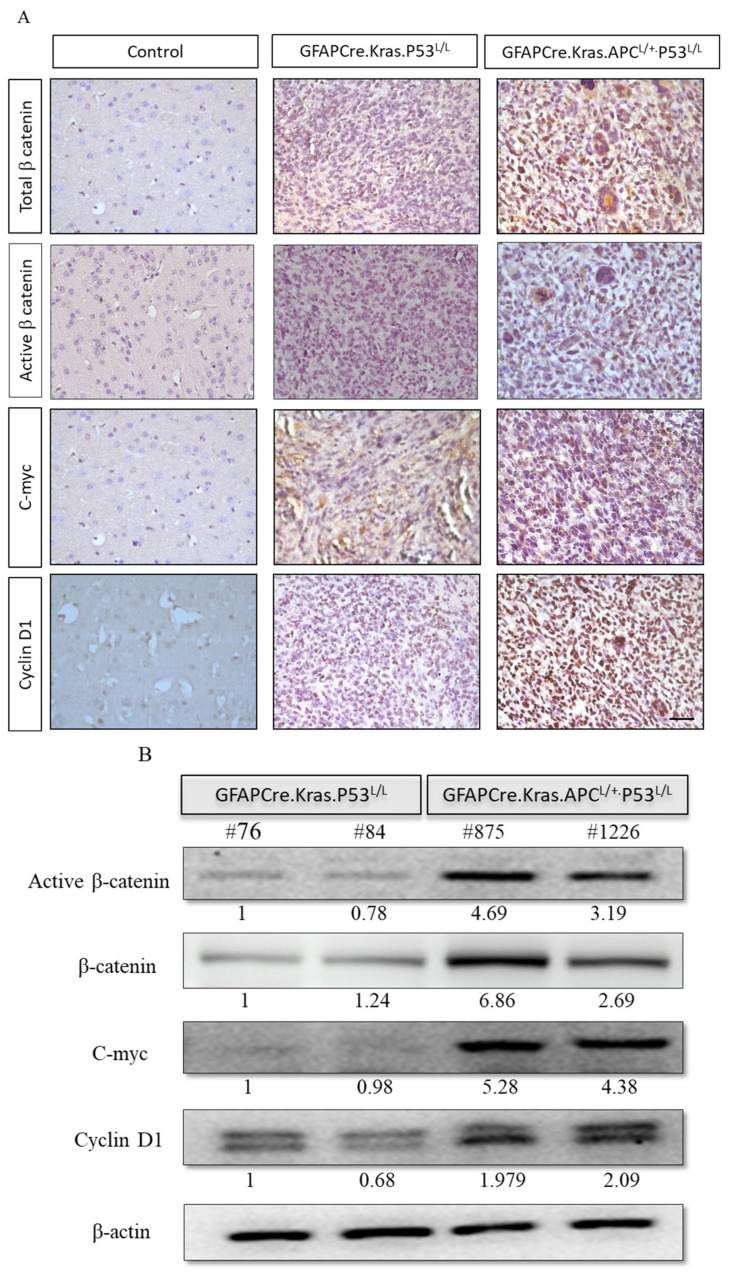
Enhanced activation of the Wnt/β-Catenin pathway in GBM from GFAP-Cre; Kras^G12D^; APC^L/+^; P53^L/L^ mice. (**A**) Immunohistochemistry analysis reveals the activation of the Wnt/β-catenin pathway, as evidenced by the high expression of β-catenin, c-myc, and cyclin D1 proteins. Bars: 100 μm. (**B**) Western blot analysis shows the expression of active β-catenin, β-catenin, c-myc, and cyclin D1 in primary GBM cells isolated from GFAP-Cre; Kras^G12D^; P53^L/L^ (#76; #84) and GFAP-Cre; Kras^G12D^; APC^L/+^; P53^L/L^ (#875; #1226) mice. β-Actin levels were used for protein loading normalization. Relative pixel intensities were measured via densitometry analysis using ImageJ 1.45s software.

**Figure 5 cancers-16-01046-f005:**
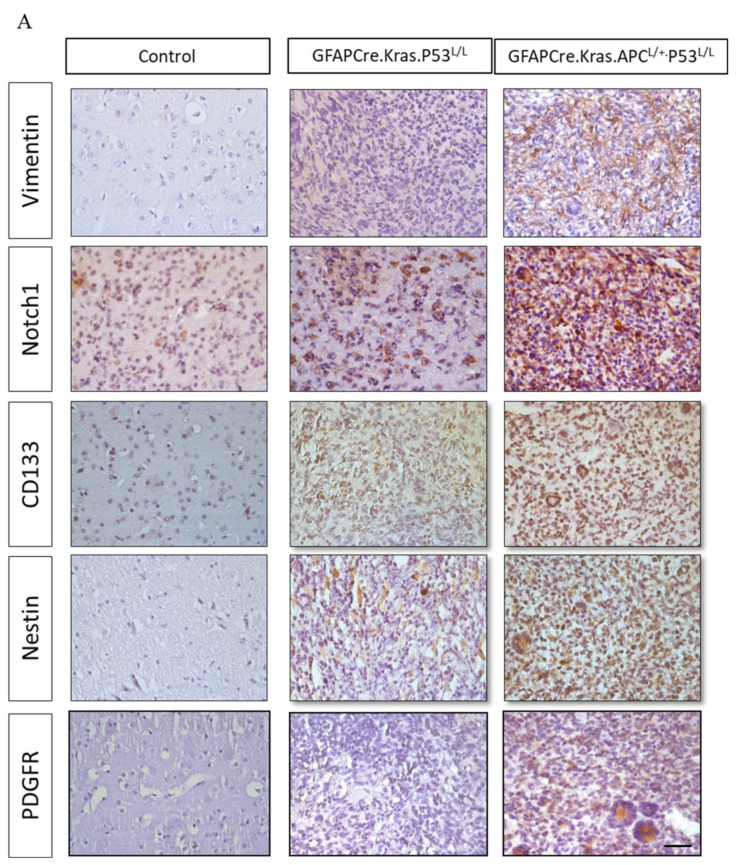
Regulation of cancer-stem-like properties in GBM mice with APC haploinsufficiency. (**A**) Immunohistochemistry analysis was performed to evaluate the expression levels of Vimentin, Notch1, CD133, Nestin, and PDGFRα proteins in formalin-fixed brain tissues obtained from GFAP-Cre; P53^L/L^, GFAP-Cre; Kras^G12D^; P53^L/L^, and GFAP-^Cre^; Kras^G12D^; APC^L/+^; P53^L/L^ mice. Bars: 100 μm. (**B**) Western blot analysis was conducted to detect the protein levels of Vimentin, GFAP, CD133, Nestin, and PDGFRα in primary GBM cells derived from GFAP-Cre; Kras^G12D^; P53^L/L^ (#76; #84) and GFAP-Cre; Kras^G12D^; APC^L/+^; P53^L/L^ (#875; #1226) mice, respectively. The expression of β-actin was used to ensure uniform protein loading across all lanes. Relative pixel intensities were measured via densitometry analysis using ImageJ 1.45s software.

**Figure 6 cancers-16-01046-f006:**
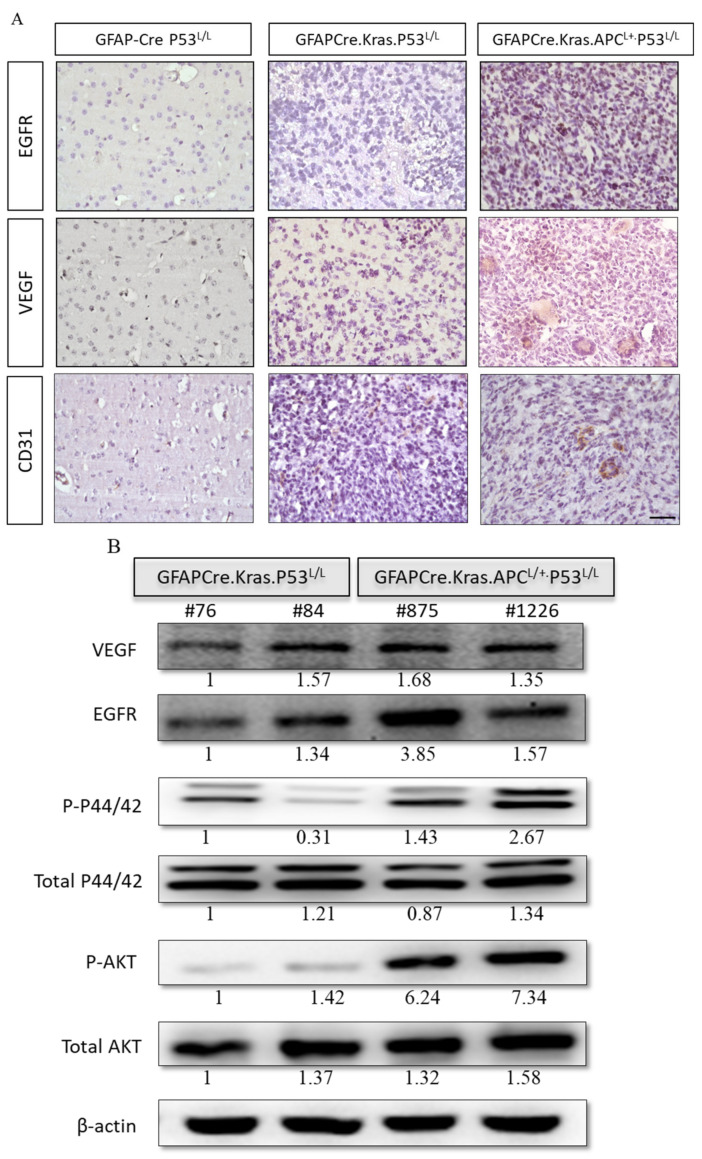
Levels of expression of the VEGF and EGFR kinase pathways in GBM derived from GFAP-Cre; P53^L/L^, GFAP-Cre; Kras^G12D^; P53^L/L^, and GFAP-Cre; Kras^G12D^; APC^L/+^; P53^L/L^ mice. (**A**) Immunohistochemical analysis of EGFR and VEGF in brain tissue sections from GFAP-Cre; P53^L/L^, GFAP-Cre; Kras^G12D^; P53^L/L^, and GFAP-Cre; Kras^G12D^; APC^L/+^; P53^L/L^ mouse brain tissues. Bars: 100 μm. (**B**) Western blot analyses assess the expression levels of EGFR, VEGF, total downstream markers p44/42, P-p44/42, P-Akt, and total Akt protein in GFAP-Cre; Kras^G12D^; P53^L/L^ (#76; #84) and GFAP-Cre; Kras^G12D;^ APC^L/+^; P53^L/L^ (#875; #1226) cells. β-Actin was used as a loading control. Relative pixel intensities were measured via densitometry analysis using ImageJ 1.45s software.

## Data Availability

Data available upon request from the corresponding author.

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
