# Peer review of "Haploinsufficiency of Adenomatous Polyposis Coli Coupled with Kirsten Rat Sarcoma Viral Oncogene Homologue Activation and P53 Loss Provokes High-Grade Glioblastoma Formation in Mice"

_cancers, 2024, doi:10.3390/cancers16051046_

Round 1
Reviewer 1 Report
Comments and Suggestions for Authors
The authors developed the GFAP-Cre/KrasG12D/APCL/+/ P53L/L transgenic mice model. This group also provides the transgenic mouse model that can promote GBM tumorigenesis, particularly when combined with the activation of the CD133, a cancer stem cell marker, EGFR and VEGF signaling pathways, and WNT/β-catenin signaling pathway. Therefore, this model may improve studies on oncogenic mechanisms or therapeutic strategies by using this model in the future.
There are some minor concerns that the authors need to clarify or further address before the paper can be re-considered for publication:
Minor comment:
1. In Figures 4 and 5, the authors should add control data in the western blot.
2. In Figure 6, the authors should add control data in IHC and western blot.
Comments on the Quality of English LanguageAlthough the general format of the paper is appropriate, minor grammatical and grammatical corrections are needed in the article; therefore, the author should seek the help of a professional editor or a native speaker to improve the English writing of this paper.
Author Response
Dear Editor,
Thank you very much for accepting our manuscript pending suggested minor revision of the article according to the reviewers’ comments. We want to extend our appreciation for taking the time and effort necessary to provide such insightful guidance. We have addresses the reviewers’ comments by providing clarifications to their queries and by incorporating modification wherever it is necessary as outlined in this letter. We hope you find these revisions an improvement.
Review 1
The authors developed the GFAP-Cre/KrasG12D/APCL/+/ P53L/L transgenic mice model. This group also provides the transgenic mouse model that can promote GBM tumorigenesis, particularly when combined with the activation of the CD133, a cancer stem cell marker, EGFR and VEGF signaling pathways, and WNT/β-catenin signaling pathway. Therefore, this model may improve studies on oncogenic mechanisms or therapeutic strategies by using this model in the future.
There are some minor concerns that the authors need to clarify or further address before the paper can be re-considered for publication:
Minor comment:
- In Figures 4 and 5, the authors should add control data in the western blot.
Response: We appreciate your earnest review and insightful comments. Due to our inability to cultivate normal mouse glial cells for use as controls, we resolved this issue by employing ImageJ software to quantitatively confirm the distinctions and present the data in this revised version.
- In Figure 6, the authors should add control data in IHC and western blot.
Response: In response to reviewer’s suggestion regarding the IHC data, we would like to clarify that GFAP-Cre; P53L/L mice were employed as the control group since they did not exhibit the development of any brain lesions. This choice was made to establish a baseline for comparison with the experimental groups. Additionally, in order to strengthen the reliability of our western blot analysis, we have conducted a semi-quantitative analysis of protein expression levels using Image J software. The results of this analysis further support and reinforce the conclusions drawn in this revised version of our study.
We greatly appreciate the reviewers for their helpful comments. We hope the revised manuscript is acceptable for publication.
With kindest regards
Kuang-hung Cheng, Ph.D
Chair and Professor of Biomedical Science Institute
National Sun Yat-Sen University
Kaohsiung, Taiwan 807
TEL: 886-7-5252000 ext 5817
Fax: 886-7-5250197
Email: khcheng@faculty.nsysu.edu.tw
Reviewer 2 Report
Comments and Suggestions for Authors
The authors investigated the role of the functions of several genes: APC haploinsufficiency, mutant KrasG12D activation and p53 loss in the induction of glioblastoma multiforme tumorigenesis on GFAP Cre transgenic mice strain. They generated GFAP-Cre/Kras G12P /APCL/+ /p53 l/l compound mice. The behavioral assessment of these mice was conducted before autopsy and histological analysis. Primary GBM cell lines were isolated from mice’s brain tumors. They were further used to perform several tests to evaluate the identification as glioblastoma cell lines so that they could be used for analysis of molecular mechanisms of GBM tumorigenesis. They found several hallmarks of human glioblastoma.
This is a very interesting study pertaining glioblastoma multiforme which is one of the deadliest tumors with very low survival rate. The 5-year survival rate of the patients is less than 10%. Glioblastoma multiforme accounts for 48,6% of the incidence of malignant brain tumors. Epidemiological statistics shows that the median survival time of patients with glioblastoma is 8 months.
Elucidation of molecular features of GBM tumors should be helpful for the future targeted therapies which would extend the survival rate.
Author Response
Dear Editor,
Thank you very much for accepting our manuscript pending suggested minor revision of the article according to the reviewers’ comments. We want to extend our appreciation for taking the time and effort necessary to provide such insightful guidance. We have addresses the reviewers’ comments by providing clarifications to their queries and by incorporating modification wherever it is necessary as outlined in this letter. We hope you find these revisions an improvement.
Review 2
The authors investigated the role of the functions of several genes: APC haploinsufficiency, mutant KrasG12D activation and p53 loss in the induction of glioblastoma multiforme tumorigenesis on GFAP Cre transgenic mice strain. They generated GFAP-Cre/Kras G12P /APCL/+ /p53 l/l compound mice. The behavioral assessment of these mice was conducted before autopsy and histological analysis. Primary GBM cell lines were isolated from mice’s brain tumors. They were further used to perform several tests to evaluate the identification as glioblastoma cell lines so that they could be used for analysis of molecular mechanisms of GBM tumorigenesis. They found several hallmarks of human glioblastoma.
This is a very interesting study pertaining glioblastoma multiforme which is one of the deadliest tumors with very low survival rate. The 5-year survival rate of the patients is less than 10%. Glioblastoma multiforme accounts for 48,6% of the incidence of malignant brain tumors. Epidemiological statistics shows that the median survival time of patients with glioblastoma is 8 months.
Elucidation of molecular features of GBM tumors should be helpful for the future targeted therapies which would extend the survival rate.
Response: We appreciate your overall assessment of our manuscript, and carefully revise this manuscript. We hope the revised manuscript is acceptable for publication.
With kindest regards
Yours Sincerely
Kuang-Hung Cheng
Kuang-hung Cheng, Ph.D
Chair and Professor of Biomedical Science Institute
National Sun Yat-Sen University
Kaohsiung, Taiwan 807
TEL: 886-7-5252000 ext 5817
Fax: 886-7-5250197
Email: khcheng@faculty.nsysu.edu.tw
Reviewer 3 Report
Comments and Suggestions for Authors
* The authors investigated the effects of APC Haploinsufficiency, Kras Activation, and P53 Loss on the pathogenesis of Glioblastoma. The manuscript is well constructed. I have some issues.
* The title must not contain any abbreviations.
* Most abbreviations in the abstract are not defined.
* Line 134: Please, group all primers in a table with their conditions.
* Line 156: Why there were parasagittal sections and coronal sections?
* Lines 159-162: are repetitive.
* Line 169, 170: why did you dry at 60~65℃ for 30 minutes?
* Line 181: Please, include all antibodies with their information in a table.
* Line 227: Please, include all antibodies with their information in a table.
* Figure 2: Please, put a ruler to define the actual size of the samples in A, use the histopathological lesions score in B, and quantitatively measure the reaction in C by image J.
* Figure 3: Please, quantitatively measure the reaction in A by image J.
* Figure 4: Please, quantitatively measure the reaction in A, B by image J.
* Figure 5: Please, quantitatively measure the reaction in A, B by image J.
* Figure 6: Please, quantitatively measure the reaction in A, B by image J.
* Please, put the original western blot figures with their protein ladder to confirm the molecular weight for every examined protein. Without protein ladder, all your western blot figures are not confirmed.
Comments on the Quality of English Language
* Your manuscript language in general is good but should be revised for minor punctuation issues by a specialized editing company.
Author Response
Dear Editors,
Thank you very much for accepting our manuscript pending suggested revisions of the article according to the reviewers’ comments. We want to extend our appreciation for taking the time and effort necessary to provide such insightful guidance. We have addresses the reviewers’ comments by providing clarifications to their queries and by incorporating modification wherever it is necessary as outlined in this letter. We hope you find these revisions an improvement.
* The title must not contain any abbreviations.
Response: Thank you for your valuable advice. We have corrected our title in the revised manuscript according to your comment.
* Most abbreviations in the abstract are not defined.
Response: We apologize for forgetting to address the abbreviations clearly. This point is well taken
* Line 134: Please, group all primers in a table with their conditions.
Response: Thank you very for your advice. I want to express my gratitude for the reviewer’s valuable advice on our manuscript. The reviewer’s input has been instrumental in refining our work. I have carefully considered the review’s suggestion, and as a result, we have added a supplementary Table S1 to consolidate all primers. This table now includes their respective conditions, providing a more organized presentation of the information.
* Line 156: Why there were parasagittal sections and coronal sections?
Response: This point is well taken. Thank you for your advice; it has been duly noted. Following a comprehensive reassessment of our mouse brain dissection and tissue fixation procedures, we have decided to omit coronal sections from our process. We appreciate your careful review of our manuscript.
* Lines 159-162: are repetitive.
Response: I appreciate the reviewer’s careful review and guidance. We have taken this point into consideration, and as suggested, we have deleted the repetitive sentences in the revised version.
* Line 169, 170: why did you dry at 60~65℃ for 30 minutes?
Respone: We appreciate your careful review of our manuscript. Dried no bake slides at 60~65℃ for 30 minutes in order to melt the paraffin wax.
* Line 181: Please, include all antibodies with their information in a table.
Response: Thank you for your valuable advice. We have carefully considered the review’s suggestion, and as a result, we have added a supplementary Table S2 to consolidate all antibodies.
* Line 227: Please, include all antibodies with their information in a table.
Response: Thank you for your valuable advice. This point is well taken.
* Figure 2: Please, put a ruler to define the actual size of the samples in A, use the histopathological lesions score in B, and quantitatively measure the reaction in C by image J.
Response: Done: We agree with your comment. We have tried our best to make new a figure 2 (including put a ruler to define the tumor sizes) that could facilitate the interpretation of the results in our revised manuscript. In Figure 2C, all murine primary GBM cells are GFAP+, eliminating the need for further quantitative measurement.
* Figure 3: Please, quantitatively measure the reaction in A by image J.
* Figure 4: Please, quantitatively measure the reaction in A, B by image J.
Response: The reviewers rightly emphasized the importance of confirming our findings between Immunohistochemistry (IHC) and western blot analyses. We acknowledge this concern and have taken steps to strengthen the correlation between these results. Our IHC data yields significant biological evidence in support of our study, and we have utilized western blot analysis to provide semi-quantitative confirmation. Furthermore, in response to the reviewer's suggestion, we have incorporated ImageJ analysis to offer a more precise quantitative measurement of protein expression levels in Figures 4, 5, and 6. These adjustments aim to reinforce the robustness of our results and ensure a comprehensive examination of the data.
* Figure 5: Please, quantitatively measure the reaction in A, B by image J.
Response: Our IHC data yields substantial biological evidence in support of our study. The western blot results serve as a semi-quantitative confirmation of our IHC findings. In response to the reviewer's suggestion, we have incorporated ImageJ analysis to quantitatively assess protein expression levels in Figures 4, 5, and 6.
* Figure 6: Please, quantitatively measure the reaction in A, B by image J.
Response: Done. Our IHC data yields substantial biological evidence in support of our study. The western blot results serve as a semi-quantitative confirmation of our IHC findings. In response to the reviewer's suggestion, we have incorporated ImageJ analysis to quantitatively assess protein expression levels in Figures 4, 5, and 6.
* Please, put the original western blot figures with their protein ladder to confirm the molecular weight for every examined protein. Without protein ladder, all your western blot figures are not confirmed.
Response: I appreciate the reviewer’s prompt response and valuable advice. The reviewer point about adding protein markers to our uncropped western blot figures is well taken. In response to the reviewer’s suggestion, we have now included the protein markers to confirm the molecular weight for every examined protein in our revised figures.
We believe these revisions address the concerns raised during the review process and improve the overall quality and reliability of our manuscript. We appreciate the time and effort invested by the reviewers and the editorial team in evaluating our work. We hope the revised manuscript is acceptable for publication.
With kindest regards
Kuang
Kuang-hung Cheng, Ph.D
Chair and Professor of Biomedical Science Institute
National Sun Yat-Sen University
Kaohsiung, Taiwan 807
TEL: 886-7-5252000 ext 5817
Fax: 886-7-5250197
Email: khcheng@faculty.nsysu.edu.tw
Reviewer 4 Report
Comments and Suggestions for Authors
In this research, to further explore the pathogenesis and progression of glioblastoma, the authors developed GFAP-Cre driven mouse model and demonstrated that activated KRAS and p53 deficiency play distinct and cooperative roles to initiate glioma tumorigenesis. And the combination of APC haplo-insufficiency with mutant Kras activation and p53 deletion resulted in the rapid progression of glioblastoma multiforme (GBM). Their GBM models closely mimic the human disease. Overall, the experimental design, data colllection/interpretation, discussion and the references are basically nice. The authors needs to check the following issues.
1. In "2.3 Behavioral Assessment of Transgenic Mice", the authors analyzed the behaviours of the experimental mice and categorized into several types, however, no related results/figures/tables was found in "3. Results".
2. check this sentence in "2.4 Histology analysis":
These tissue sections were embedded in paraffin, cut into 5μm-thick sections, and stained with hematoxylin and eosin for histological analysis, were sacrificed, and their brains were fixed in 10% buffered formalin acetate overnight, followed by fixation in 75% ethanol.
2. In "2.8 Cell proliferation assay", the MTT method basically is for the cell viability, not for the cell proliferation.
3. In Fig 2B, what's the exact parts/spots of these sections in mouse brain?
4. How the brains of the experimental mice were taken? Did the authors used the transcardial perfusion with saline and 4% ice-cold PFA in PBS? If not, the remaining blood in the brain tissues may influence the fluoresent staining.
Comments on the Quality of English LanguageThe manuscript is written in fluent English, but minor check is need throughout.
Author Response
Date: 11/18/2023
Dear Editor,
Thank you very much for accepting our manuscript pending suggested minor revision of the article according to the reviewers’ comments. We want to extend our appreciation for taking the time and effort necessary to provide such insightful guidance. We have addresses the reviewers’ comments by providing clarifications to their queries and by incorporating modification wherever it is necessary as outlined in this letter. We hope you find these revisions an improvement.
Review 03
In this research, to further explore the pathogenesis and progression of glioblastoma, the authors developed GFAP-Cre driven mouse model and demonstrated that activated KRAS and p53 deficiency play distinct and cooperative roles to initiate glioma tumorigenesis. And the combination of APC haplo-insufficiency with mutant Kras activation and p53 deletion resulted in the rapid progression of glioblastoma multiforme (GBM). Their GBM models closely mimic the human disease. Overall, the experimental design, data colllection/interpretation, discussion and the references are basically nice. The authors need to check the following issues.
Response: We thank the reviewer for the compliments and especially for the constructive suggestions.
- In "2.3 Behavioral Assessment of Transgenic Mice", the authors analyzed the behaviours of the experimental mice and categorized into several types, however, no related results/figures/tables was found in "3. Results".
Response: Thank you for the constructive suggestion from the reviewer. In response, we have included a Supplemental Table S3 to present the representative results of our behavioral testing for these GBM mice.
- check this sentence in "2.4 Histology analysis":
These tissue sections were embedded in paraffin, cut into 5μm-thick sections, and stained with hematoxylin and eosin for histological analysis, were sacrificed, and their brains were fixed in 10% buffered formalin acetate overnight, followed by fixation in 75% ethanol.
Response: We would like to thank you for providing your constructive and detailed review comments on our manuscript. This point is well taken.
- In "2.8 Cell proliferation assay", the MTT method basically is for the cell viability, not for the cell proliferation.
Response: This point is well taken.
- In Fig 2B, what's the exact parts/spots of these sections in mouse brain?
Response: These sections were dissected from the frontal lobes of the cerebral cortex. We added this information in our revised version. Meanwhile, our GBM mice developed multifocal, infiltrating glioma most located in the front or temporal lobes of cerebral cortex and thalamus near lateral ventricle (LV) with 100% penetrance.
- How the brains of the experimental mice were taken? Did the authors used the transcardial perfusion with saline and 4% ice-cold PFA in PBS? If not, the remaining blood in the brain tissues may influence the fluoresent staining.
Response: We appreciate your thorough review and your comments regarding our methodology, particularly the absence of transcardial perfusion in our brain tissue collection process. We would like to provide further details to address your concerns.
In our study, mice were euthanized with an intraperitoneal injection of Tribromoethanol (Avertin) at a dose of 250 mg/kg. Subsequently, cervical dislocation was performed, and whole brains were promptly dissected. We acknowledge that transcardial perfusion is a widely-recognized technique for removing blood and fixing tissues, but we deliberately chose an alternative approach based on our confident experimental skills and specific considerations related to our study objectives.
Following brain dissection, we immediately immersed the tissues in 10% formalin for overnight fixation, followed by additional fixation in 70% ethanol before embedding. This method was chosen to efficiently preserve the tissue structure and molecular components relevant to our experimental goals. Additionally, in response to concerns about the potential influence of fluorescent staining, we implemented a negative control strategy. Specifically, we utilized normal mouse brains as negative controls to account for any non-specific fluorescence or background signal.
We greatly appreciate the reviewers for their helpful comments. We hope the revised manuscript is acceptable for publication.
With kindest regards
Yours Sincerely
Kuang-Hung Cheng
Kuang-hung Cheng, Ph.D
Chair and Professor of Biomedical Science Institute
National Sun Yat-Sen University
Kaohsiung, Taiwan 807
TEL: 886-7-5252000 ext 5817
Fax: 886-7-5250197
Email: khcheng@faculty.nsysu.edu.tw
Round 2
Reviewer 3 Report
Comments and Suggestions for Authors
* Some comments have not been responded to fully.
* The abstract still has some non-defined abbreviations such as KRAS, and APC.
* Figure 3: Please, quantitatively measure the reaction in A by image J. Not responded.
* Figure 4 : beta catenin have many bands. I suggest to repeat.
* Figure 5: I suggest to repeat Nestin and Cd133.
* How did the authors add the protein markers? Protein ladder must put during the western blot experiment? how did you identify the molecular weights of the proteins without protein ladders?
Author Response
Date: 01/20/2024
Dear reviewer,
We express our gratitude for accepting our manuscript, subject to minor revision in accordance with the constructive feedback from the reviewer. We appreciate the time and effort you and the reviewers invested in providing insightful guidance.
In response to the reviewer's comments, we have diligently addressed the suggested revisions. Kindly find the details below:
* The abstract still has some non-defined abbreviations such as KRAS, and APC.
Reply: Done. We have well defined the abbreviations in the abstract.
* Figure 3: Please, quantitatively measure the reaction in A by image J. Not responded.
Reply: We have successfully addressed the reviewers' comments and made the necessary revisions to the manuscript. Specifically, we have incorporated quantitative measurements for TUNEL, Ki67, and p-Histone3 IHC in Figure 3A.
* Figure 4 : beta catenin have many bands. I suggest to repeat.
Reply: we redo the western blot for b-catenin in our revision manuscript.
* Figure 5: I suggest to repeat Nestin and Cd133.
Reply: yes, we have reconfirmed the Western blot results for Nestin and CD133, and we are pleased to report that we obtained similar results. This reaffirms the robustness and consistency of our findings.
* How did the authors add the protein markers? Protein ladder must put during the western blot experiment? how did you identify the molecular weights of the proteins without protein ladders?
Reply: Thank you for your understanding, and we acknowledge the importance of presenting original Western blot images with standard markers as part of the review process. However, we are currently facing challenges in retrieving the original Western blot images, specifically those containing standard molecule markers, from our students' archives.
The main obstacle arises from the fact that the students who conducted these experiments have graduated several years ago, making it difficult to access the original files. In our laboratory, we typically identify the molecular weights of proteins without protein ladders by cutting the PVDF membrane into three pieces, with divisions at 100 kDa and 35 kDa.
To address this concern, we are considering presenting the original Western blot images that have been appropriately cropped from our supplementary data. We believe this approach will provide a clear representation of our results while accommodating the challenges we currently face.
We appreciate your understanding in this matter.
We believe these revisions address the concerns raised during the review process and improve the overall quality and reliability of our manuscript. We appreciate the time and effort invested by the reviewers and the editorial team in evaluating our work. We hope the revised manuscript is acceptable for publication.
With kindest regards
Best
Kuang
1/20/2024

Round 3
Reviewer 3 Report
Comments and Suggestions for Authors
* The protein ladder has not been solved even for the new western blot ones.
* The authors did not show the new original western blot images.
* The authors did not reveal the quantative measurment for GFAP in fig. 2.
* The Relative pixel intensities of some western blot figures did not correctly measured such as C-myc, GFAP, CD133, PDGFRalpha, total AKT.